



An investigation on hygroscopic properties of 15 black carbon
(BC) from different carbon sources: Roles of organic and
inorganic components
Minli Wang[a], Yiqun Chen[a], Heyun Fu[a], Xiaolei Qu[a], Bengang Li[b], Shu Tao[b], and
Dongqiang Zhu[a,b,*]
[a] State Key Laboratory of Pollution Control and Resource Reuse, School of the
Environment, Nanjing University, Jiangsu 210046, China
[b] School of Urban and Environmental Sciences, Key Laboratory of the Ministry of
Education for Earth Surface Processes, Peking University, Beijing 100871, China
[*]Corresponding author. Tel.: +86 010-62766405. E-mail: zhud@pku.edu.cn (D. Zhu)



**Abstract**
The hygroscopic behavior of black carbon (BC) has a significant impact on global
and regional climate change. However, the mechanism and factors controlling the
hygroscopicity of BC from different carbon sources are not well understood. Here, we
systematically measured the equilibrium and kinetics of water uptake by 15 different
BC (10 herb-derived BC, 2 wood-derived BC, and 3 soot) using gravimetric water
vapor sorption method combined with in-situ diffuse reflectance infrared Fourier
transform spectroscopy (DRIFTS). In the gravimetric analysis, the sorption/desorption
equilibrium isotherms were measured under continuous-stepwise water vapor pressure
conditions, while the kinetics was measured at a variety of humidity levels obtained
by different saturated aqueous salt solutions. The equilibrium water uptake of the BC
pool at high relative humidity (> 80%) positively correlated to the dissolved mineral
content (0.01–13.0wt%) ($R^2 = 0.86$, $P = 0.0001$) as well as the content of the
thermogravimetrically analyzed organic carbon (OC$_{TGA}$, 4.48–15.25wt%) ($R^2 = 0.52$,
$P = 0.002$) and the alkali-extracted organic carbon (OC$_{AE}$, 0.14–8.39wt%) ($R^2 = 0.80$,
$P = 0.0001$). In contrast, no positive correlation was obtained with the content of total
organic carbon or elemental carbon. Among the major soluble ionic constituents,
chloride and ammonium were each correlated with the equilibrium water uptake at
high relative humidity. Compared with the herbal BC and soot, the woody BC had
much lower equilibrium water uptake, especially at high relative humidity, likely due
to the very low dissolved material content and OC content. The DRIFTS analysis
provided generally consistent results at low relative humidity. The kinetics of water
uptake (measured by pseudo-second order rate constant) correlated to the content of
OC$_{TGA}$ and OC$_{AE}$ as well as the content of chloride and ammonium at low relative
humidity (33%), but to the porosity of bulk BC at high relative humidity (94%). This





was the first study to show that BC of different types and sources has greatly varying
hygroscopic properties.

## 1. Introduction

Black carbon (BC) refers to a collective term of recalcitrant carbonaceous materials
generated from incomplete combustion of biomass and fossil fuels (Bond et al., 2013).
BC is ubiquitous in the atmosphere and is a major component of atmospheric
carbonaceous aerosols (Schwarz et al., 2008). Due to the strong ability to absorb
visible light (Yuan et al., 2015), BC causes positive radiative forcing effects on
climate and is considered an important factor driving global warming (Matthews et al.,
2009). Once immersed into cloud droplets, BC can also facilitate water evaporation
and cloud dispersion via enhanced absorption of solar radiation, and thus produce
indirect radiative forcing effects (Powelson et al., 2014). Additionally, the large
specific surface area of BC creates a potential for heterogeneous reactions with trace
gases (such as volatile halocarbons) in the atmosphere (Qiu et al., 2012), therefore
heavily impacting atmospheric chemistry and air quality. Hygroscopicity is a key
determinant of physical, chemical, and optical properties of BC by changing particle
size, phase state, and quality and morphological development, which in turn affect
aerosol radiation effect, formation of cloud and ice nuclei, and
heterogeneous chemical reactions (Bond et al., 2013; Liu et al., 2018). Furthermore,
the hygroscopicity of BC is an important factor contributing to the risk of human
respiratory infections, cardiovascular diseases, and other infectious diseases (Haddrell
et al., 2015).
BC is composed of a complex matrix of inorganic and organic components. The
inorganic components consist of a variety of amorphous or crystalline salts (sulfates,
chlorides, etc.) as well as semi-crystalline minerals (such as silica) (Stanislav et al.,


2013). Despite the relatively low content in BC, the inorganic components play a
significant role in water uptake of BC, depending on their types, contents, and mixing
ratios (Lewis et al., 2009). As the factors and processes governing the hygroscopic
deliquescence of inorganic salts are very complicated (Reid et al., 2005; Zhang et al.,
2012), it is of a great challenge to assess the contribution of a specific salt to the
overall hygroscopicity of BC, and thus, its role is still controversial. Previous studies
suggested that KCl was responsible for the high hygroscopicity of BC produced by
fresh biomass burning (Posfai et al., 2003), while the presence of $K_2SO_4$ or $KNO_3$
caused the low hygroscopicity of BC produced by aged biomass burning (Li et al.,

73 2003).

The organic components in BC consist of graphitized elemental carbon (EC) and
non-condensed, amorphous organic carbon (OC) (Lian and Xing, 2017). The
contribution of EC to the overall hygroscopicity of BC is considered low due to the
very high hydrophobicity (Seisel et al., 2005). The role of OC in the hygroscopic
growth of BC is intricate and debatable. The positive effect of OC is mainly attributed
to water absorption by the oxygen-containing functional groups (Fletcher et al., 2007;
Suda et al., 2014). The negative effect of OC is suggested to stem from the impeded
mass transfer process of water molecules by formation of coatings on hygroscopic
minerals or inhomogeneous morphology inside the particle (Sjogren et al., 2007;
Stemmler et al., 2008). In addition to the total content, the molecular weight, water
solubility, surface tension, and type and content of functional groups of OC were all
found to influence the overall hygroscopicity of BC. Moreover, the effect of OC on
BC hygroscopicity is further complicated by the formation of organic minerals
(presumably through strong covalent bonds) (Archanjo et al., 2014; Reid et al., 2005;
Zuend et al., 2011).



The carbon sources for BC particles released into the atmosphere are expected to be
highly diversified and cover a wide range of plant biomass, coals, and refined oil
products, although their quota can hardly be accurately assessed (Andreae and
Gelencser, 2006). The chemical compositional and structural properties of BC depend
significantly on carbon sources and combustion conditions (Xiao et al., 2018). For
instance, crop residue-derived biochar often has higher mineral content than
wood-derived charcoal, while biochar formed at higher pyrolysis temperatures
generally have higher aromaticity, specific surface area, and pore volume but lower
polarity than biochar formed at lower temperatures (Wei et al., 2019). Previous studies
on hygroscopicity of BC have mainly focused on wood-derived BC (Carrico et al.,
2010; Day et al., 2006), whereas BC from other carbon sources has been largely
overlooked. It remains unclear whether BC from different carbon sources would differ
significantly in hygroscopicity.
Herein we systematically investigated the equilibrium and kinetics of water uptake
by 15 different BC samples derived from wood, herb, coal, and diesel at varying
relative humidity (RH) levels by gravimetric sorption and in-situ diffuse reflectance
infrared Fourier transform spectroscopy (DRIFTS). The chemical, compositional, and
structural properties of the tested BC pool were thoroughly characterized to unveil the
key factors controlling the hygroscopic properties.
**2. Experimental methods**
**2.1. Preparation of BC**
A total of 15 BC samples were tested, including 10 herb-derived BC, 2 wood-derived
BC, and 3 soot. The herbal BC from amaranth, peanuts, pea, grass, rice, wheat, corn,
millet, sorghum, and bamboo, and woody BC from red pine and poplar, respectively
were prepared by pyrolysis. Briefly, the dried and dehydrated biomass was pulverized


into a fine powder using a high-speed pulverizer (FW 100, Tianjin Taisite Instrument,
China), and pyrolyzed in a muffle furnace under an oxygen-limited conditions. The
oven temperature was programmed to increase from 20 to 400 °C in 2 h and
maintained at 400 °C for 3 h. The Household soot was freshly collected on the inner
wall of the stove chamber produced by burning of coal and wood for winter cooking
and heating (Linkou County, Heilongjiang Province, China). The Weifu diesel soot
produced by burning diesel (# 5, China) at 1000 °C was collected by a diesel
particulate filter from the exhaust stream at a carbon deposition temperature of 250 °C.
The diesel engine soot was taken from the freshly discharged exhaust particles on the
tailpipe of a diesel truck (# 0, 3.7 L, CY4100, Dongfeng, China). The obtained BC
and soot samples were further ground to pass a 100-mesh sieve (0.15 mm) and stored
sealed in a brown glass bottle at 4 °C.
**2.2. Characterization of BC**
Elemental analysis (EA) was performed using a Vario micro cube elemental analyzer
(Elementar, Hanau, Germany). Surface elemental compositions were measured by
X-ray photoelectron spectroscopy (XPS) (PHI 5000 VersaProbe, UlVAC-PHI, Japan).
Mineral compositions were measured by X-ray fluorescence (XRF) (ARL-9800, ARL
Corporation, Switzerland). Fourier-transform infrared (FTIR) spectra were recorded
on a Bruker Tensor 27 Karlsruhe spectrometer (Germany) using KBr pellets in the
range of 400 to 4000 cm$^{-1}$. X-ray diffraction (XRD) spectra were recorded on an AXS
D8 Advance spectrometer (Germany) using Cu Kα radiation at a 2θ angle ranging
from 5 to 70°. Raman spectra were collected on a Horiba Jobin Yvon LabRam
HR-800 spectrometer equipped with a 514 nm laser (France). $N_2$ adsorption isotherms
to the 15 BC were obtained on a Micrometrics ASAP 2020 (Micromeritics Instrument
Co., Norcross, GA, USA) apparatus at -196 °C (77 K).





Three different methods, thermogravimetric analysis (TGA), alkali extraction, and
water extraction, were explored to quantify the content of OC in BC (referred to as
$OC_{TGA}$, $OC_{AE}$, and $OC_{WE}$, respectively). The content of $OC_{TGA}$ was measured as the
weight loss during the heating of BC from 30 to 300 °C at a ramp of 10 °C per minute
in a nitrogen flow (Han et al., 2013) using a TGA 8000 analyzer (PerkinElmer, USA).
To measure the content of $OC_{AE}$, the BC sample was mixed with 0.1 M NaOH at a
solid-to-solution ratio of 1: 10 (w/w) and magnetically stirred for 12 h, followed by
filtration through a 0.45-μm filter membrane (Pall, USA) (Song et al., 2002). The
procedure was repeated until the filtered supernatant was colorless. The filtrate was
collected and the total organic carbon (TOC) content was measured by a TOC
analyzer (TOC-5000A, Shimadzu, Japan). For three selected BC (Grass BC, Wheat
BC and Household soot), the filtrate was precipitated by acidification (pH 1.0 with 6
M HCl), which was separated by centrifugation and dialyzed in deionized (DI) water
by dialysis bag (500 Da, Union Carbide, USA) until no chloride ion was
detected by $AgNO_3$, and then freeze-dried. The elemental compositions of the three
prepared $OC_{AE}$ were measured by EA. To measure the content of $OC_{WE}$ and dissolved
minerals, the BC suspended in DI water (BC-to-water ratio of 1: 10, w/w) was
sonicated in a water bath for several minutes, and the mixture was filtered through a
0.45-μm membrane. This procedure was repeated for 6 times. The filtrate was
collected and subjected to TOC analysis to obtain the content of $OC_{WE}$. The
concentrations of ionic constituents ($Cl^-$,$NO_3^-$,$PO_4^{3-}$,$SO_4^{2-}$,$F^-$,$COO^-$,$C_2O_4^{2-}$,
$Na^+$,$NH_4^+$,$K^+$,$Mg^{2+}$,$Ca^{2+}$,$Al^{3+}$) in the filtrate were measured using a
Dionex ICS-1100 ion chromatography (Thermo Scientific, USA). The cations were
eluted using 20 mM methanesulfonic acid on a Dionex IonPac CS12A column (4 ×
250 mm), while the anions were eluted using an eluent of 4.5 mM $Na_2CO_3$ and 0.81



mM $NaHCO_3$ on a Dionex IonPac AS14A column (4 × 250 mm). The filtrate was
further freeze-dried and baked at 600 °C for 6 h to remove organic components. The
remaining ash was weighed to determine the content of dissolved minerals in BC. A
portion of the ash was extracted three times using DI water at a solid-to-solution ratio
of 1:10 (w/w) under sonication, and the salinity of the extract was measured by a
ST3100C conductivity meter (OHAUS, USA). All reagents and chemicals used
were of analytical reagent grade.
**2.3. Measurement of BC hygroscopicity**
The hygroscopicity of BC at varying RH was measured by gravimetric method
combined with in-situ DRIFTS. The water vapor sorption/desorption isotherms to BC
under a range of continuous-stepwise water vapor pressures were acquired on
a 3H-2000 PW Multi-stations Gravimetric Method Steam Adsorption Instrument
(Beijing, China) at 25 °C using an approach similar to that in previous studies (Gu et
al., 2017). The instrument consists of two main parts: a balance chamber to determine
the sample mass to an accuracy ±1 μg and a humidity chamber to regulate the water
vapor pressure to the desired value as monitored online by a pressure sensor. Prior to
testing, the BC sample (about 10 mg) was dried at 70 °C under vacuum for 12 h to
remove pre-adsorbed gases. The amount of water sorbed to BC was monitored as the
mass difference before and after sorption. The amount of water sorbed to the sample
tube was negligible (< 0.05% of the amount of water sorbed to BC). The water
vapor pressures ranging from 10 to 94% RH were applied to the sorption isotherm
branch in a stepwise increasing sequence and to the desorption isotherm branch in a
stepwise decreasing sequence.
The kinetics of water sorption to BC was measured on a 100 mm closed quartz
chamber (Jiangsu Province, China) using a gravimetric method similar to that in


previous studies (Yuan et al., 2014). Approximately 100 mg of BC sample was dried
at 70 °C under vacuum for 12 h, weighed in a 10-mL beaker, and placed in a chamber
under controlled humidity conditions based on different saturated aqueous salt
solutions according to ASTM E104-02 (2007). The saturated solutions of $CH_3COOK$,
$MgCl_2$, $K_2CO_3$, $LiNO_3$, NaCl, KCl, and $KNO_3$ provided RH of 23%, 33%, 43%, 47%,
75%, 84%, 94%, respectively at 25 °C. The sample was continuously weighed and
recorded over a period of time (48 h for low humidity and 96 h for high humidity) to
monitor the amount of sorbed water. The RH was monitored in real time using a
Honeywell HIH4000 hygrometer (USA) with measurement variance was less than 5%.
Sorption equilibrium was reached in the late stage of the experiment as evidenced by
the stabilized constant value of sample mass. In addition to kinetic data, sorption
isotherms were also collected for the seven selected RH levels using the measured
mass under equilibrium conditions.
BC samples equilibrated at different RH levels were characterized by in situ
DRIFTS using a Bruker Tensor 27 spectrometer equipped with a high-sensitivity
mercury-cadmium-telluride (MCT) detector working under liquid $N_2$ conditions and a
chamber fitted with ZnSe windows (Harrick Scientific, USA). About 10 mg of BC
pre-dried at 70 °C under vacuum for 12 h was transferred to the chamber which was
connected to a gas feeding system. The chamber was sealed and purged with
high-purity $N_2$ at a flow rate of 100 mL per minute for at least 3 h to remove
pre-adsorbed gases on BC and to minimize the interference of environmental $CO_2$.
The humidity in the chamber was regulated by mixing high-purity $N_2$ and saturated
water vapor at 25 °C with varying ratios and monitored in real time by a hygrometer
(Vaisala Humitter, Australia). The sample was equilibrated with the gas mixture in the
chamber for at least 30 minutes to reach sorption equilibrium based on pre-determined



kinetics. The spectra were acquired by co-adding and averaging a plurality of 500
scans with a resolution of 4 cm$^{-1}$ (Song and Boily, 2013). The amount of water sorbed
to BC was monitored by the integrated intensity of the O-H stretching region from
2750 to 3660 cm$^{-1}$ (Ghorai et al., 2011).
**3. Results and discussion**
**3.1. Characteristics of BC**
Bulk elemental compositions by EA and surface elemental compositions by XPS are
summarized in Table S1. The bulk elemental compositions of all BC samples were
dominated by C and O, together accounting for 54%–96% of the total. However, the
bulk C, O compositions differed significantly among the 15 BC, ranging from 32 to
76% for C and from 16 to 69% for O. With the exception of the woody BC, the
differences were apparent within each category of the herbal BC and the soot. The
surface elemental compositions were also dominated by C and O, but the
compositional differences among the 15 BC were much smaller than the bulk
elemental compositions. Besides C and O, EA detected low amounts of N (< 3.7%)
and S (< 1.8%), and XPS detected low amounts of N (< 4.3%), Si (< 5.6%), and S (<
0.6%). The contents of oxygen-containing groups in the 15 BC were qualitatively
compared by the FTIR spectra (Figure S1). All the tested BC except Weifu diesel soot
showed characteristic peaks of esters (1700 cm$^{-1}$), ketones (1613, 1100 cm$^{-1}$), and
phenols (1270 cm$^{-1}$) (Keiluweit et al., 2010), generally with larger peak intensities
observed for herbal-derived BC and household soot.





Table 1. Chemical, compositional, and pore properties of different BC.

| Samples | OC | | | EC[d] | Dissolved minerals | Total Porosity[e] | SSA[f] |
|---|---|---|---|---|---|---|---|
| | $OC_{TGA}$[a] (wt%) | $OC_{AE}$[b] (wt%) | $OC_{WE}$[c] (wt%) | (wt%) | (wt%) | ($m^3\ g^{-1}$) | ($m^2\ g^{-1}$) |
| Amaranth BC | 6.24 | 2.6 | 1.75 | 25.84 | 10.8 | 0.004 | 0.314 |
| Grass BC | 7.24 | 2.37 | 1.01 | 51.56 | 4.8 | 0.008 | 5.587 |
| Peanuts BC | 7.45 | 1.78 | 0.8 | 41.86 | 4.2 | 0.002 | 0.192 |
| Pea BC | 9.59 | 1.98 | 0.09 | 54.48 | 3.6 | 0.005 | 4.679 |
| Rice BC | 6.81 | 0.6 | 0.11 | 48.06 | 0.6 | 0.023 | 31.88 |
| Wheat BC | 8.25 | 1.82 | 0.37 | 42.65 | 5.8 | 0.01 | 7.382 |
| Millet BC | 9.41 | 1.97 | 0.93 | 32.25 | 8 | 0.023 | 8.319 |
| Corn BC | 6.55 | 0.14 | 0.32 | 46.47 | 1.8 | 0.028 | 28.6 |
| Sorghum BC | 9.09 | 1.12 | 0.62 | 55.26 | 4.8 | 0.001 | 0.192 |
| Bamboo BC | 6.84 | 0.23 | 0.12 | 61.7 | 0.6 | 0.029 | 51.94 |
| Red pine BC | 7.4 | 0.2 | 0.05 | 62.59 | 0.01 | 0.032 | 64.24 |
| Poplar BC | 7.58 | 0.19 | 0.09 | 64.22 | 0.6 | 0.071 | 107.6 |
| Diesel engine soot | 9.57 | 1.4 | 0.78 | 27.37 | 3.6 | 0.021 | 6.119 |
| Weifu diesel soot | 4.48 | 0.57 | 0.13 | 71.98 | 3.4 | 0.484 | 194.6 |
| Household soot | 15.25 | 8.39 | 2.24 | 21.83 | 13 | 0.012 | 7.79 |

[a]Content of organic carbon determined by TGA. [b]Content of alkali-extractedorganic
carbondetermined by TOC analysis. [c]Content of water-extracted organic
carbondetermined by TOC analysis. [d]Determined by subtracting $OC_{TGA}$ content from
total organic carbon content by EA. [e]Total pore volume determined by $N_2$ adsorption
at 0.97 atmosphere pressure. [f]Specific surface area determined by the BET method.
Table 1 summarizes the contents of OC ($OC_{TGA}$, $OC_{AE}$, and $OC_{WE}$) of the 15 BC by
TGA, alkali extraction, and water extraction, respectively. For a given BC, the
contents of the three types of OC differed pronouncedly, with an increasing order of
$OC_{WE} < OC_{AE} < OC_{TGA}$. The OC content also differed within the tested BC pool,
ranging from 0.05 to 2.24wt% for $OC_{WE}$, from 0.14 to 8.39wt% for $OC_{AE}$, and from
4.48 to 15.25wt% for $OC_{TGA}$. Compared with the EC (graphitized carbon), the three
types of OC are non-condensed, amorphous, and more rich in oxygen-containing
functional groups. This was evidenced by the fact that the $OC_{AE}$ from the three
selected BC (Grass BC, Wheat BC and Household soot) had markedly higher bulk



compositions of O (results presented in Table S2). The content of EC in BC was
roughly assessed by subtracting the $OC_{TGA}$ content from the total organic carbon
content measured by EA (results presented in Table 1). The calculated EC content
negatively correlated with the $OC_{AE}$ content ($R^2 = 0.43$, $P = 0.0079$) for the examined
BC pool. This was reasonable as EC was comprised of mature, thermodynamically
stable graphitized carbons, while OC was comprised of the less mature and less
aromatic constituents remaining after pyrolysis. Except for Weifu diesel soot, the two
woody BC had the highest EC, but the lowest $OC_{AE}$ and $OC_{WE}$ among the 15 BC.
The relative abundance of EC in BC was also assessed by Raman spectroscopy
(Figure S2). The spectra of all the tested BC were dominated by a D band at 1340
$cm^{-1}$ and a G band at 1580 $cm^{-1}$, which were ascribed to carbon network defects and
the $E_{2g}$ mode of the graphitized carbon, respectively (Pimenta et al., 2007). Thus, the
ratio of these two bands ($I_D/I_G$) was inversely proportional to the in-plane crystallite
size of graphitized carbons of BC (Cancado et al., 2006). The $I_D/I_G$ ratio of the woody
BC (0.51–0.59) was less than those of the herbal BC (0.88–1.09) and the soot
(0.77–1.12) (Table S3), suggesting larger sizes of graphitized carbons in the woody
BC. This was consistent with the results of OC compositions.
The contents of dissolved minerals of the 15 BC are listed in Table 1, and their
salinities in water extracts are listed in Table S4. The two woody BC had the lowest
contents of dissolved minerals and salinities, while these contents in herbal BC and
soot were higher and varied greatly. The mineral compositions characterized by XRF
are listed in Table S5. Si-, K-rich minerals were the two major inorganic constituents
in the herbal BC and woody BC. Moreover, these two types of BC generally
contained trace amounts of S-, Cl-, Ca-, P-, Mg-, Na-, Fe-, and Al-minerals, with
lower contents observed for the woody BC. The three soot had very different mineral





compositions. Household soot was dominated by S-, Ca-, Si, and Cl-minerals, Diesel
engine soot was dominated by S-, Ca-, and Fe-minerals, while Weifu diesel soot
contained negligible mineral compositions. As reflected by the observed characteristic
peaks and associated peak intensities in the XRD spectra (Figure S3), the herbal BC
and Household soot contained more mineral species with higher contents than other
BC, whereas the two woody BC and Weifu diesel soot contained the least species and
contents of minerals. Potassium salts, amorphous silica, and sulfates were the major
minerals in the herbal BC. Soot had the largest content of sulfates among the tested
BC. According to the ion chromatograph analysis (results presented in Figure S4 and
Table S6), the major water-extracted cationic species from the tested BC were $NH_4^+$,
$K^+$, and $Ca^{2+}$, and the major anionic species were $SO_4^{2-}$, $Cl^-$, and $C_2O_4^{2-}$. The herbal
BC had high contents of $K^+$, $C_2O_4^{2-}$, and $Cl^-$, while the soot had high content of $SO_4^{2-}$.

The Brunauer–Emmett–Teller (BET) specific surface area and total porosity

measured by $N_2$ adsorption are also summarized in Table 1. A huge disparity of
specific surface area was shown among the 15 BC and among the BC within each
category, ranging from 6 to 200 $m^2$ $g^{-1}$ for the soot, from 60 to 110 $m^2$ $g^{-1}$ for the
woody BC, and from 0.1 to 52 $m^2$ $g^{-1}$ for the herbal BC. The herbal BC and woody
BC were dominated by micropores (pore size < 2 nm), which accounted for more than
50% of the total pore volume. Alternatively, mesopores (50 nm > pore size > 2 nm)
were the main pore structure of the soot, accounting for more than 61% of the total
pore volume.
**3.2. Hygroscopic properties of BC**
**Equilibrium water uptake**. Figure 1 displays sorption and desorption isotherms of
water vapor with BC plotted as equilibrium water uptake (mg $g^{-1}$) by unit mass of BC
under continuous-stepwise water vapor pressure conditions. Figure S5 displays the





equilibrium sorption isotherms at selected humidity levels obtained by using saturated
aqueous salt solutions. Under similar humidity conditions (80% and 84%), the water
uptake by the 15 BC was very close between these two humidity regulation methods
(Figure S6), reflecting their technical validity. The woody BC showed very different
sorption isotherm patterns from the herbal BC and soot. First, the water sorbing ability
of the woody BC was much lower. The maximum water uptake observed at the
highest RH (94%) was approximately 65 mg g$^{-1}$ by the woody BC, but was more than
400 mg g$^{-1}$ for the strongest sorbing herbal BC and soot. Second, much larger
water-uptake disparities were observed within the herbal BC group and the soot group
than within the woody BC group. Additionally, over the examined RH range
(10–94%), the water uptake by the woody BC increased slowly and linearly with the
RH; however, for the herbal BC and soot, the water uptake increased more rapidly
with the RH, especially under high humidity conditions (RH > 70%).

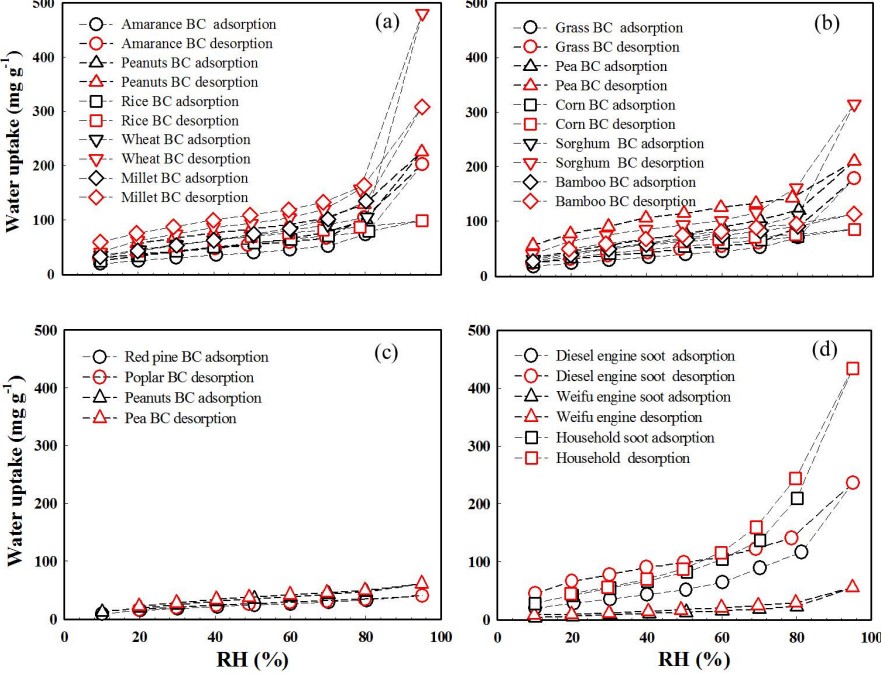






Figure 1. Sorption and desorption isotherms of water vapor plotted as water uptake
(mg g$^{-1}$) vs. relative humidity (RH, %) at equilibrium for different BC. (a) Subgroup 1
of herbal BC. (b) Subgroup 2 of herbal BC. (c) Woody BC. (d) Soot.
To better understand the underlying mechanisms and factors controlling the
hygroscopic properties of BC, linear relationships were built between the equilibrium
water uptake and a wide variety of compositional and pore property parameters of the
whole BC pool. Figure 2 displays regression relationships with contents of OC$_{TGA}$,
OC$_{AE}$, EC, dissolved minerals, major ionic species (NH$_4^+$, Cl$^-$, SO$_4^{2-}$, and C$_2$O$_4^{2-}$), and
total porosity, respectively at 94% RH. The regression relationships at 23% RH were
presented in Figure S7. The regression accuracy ($R^2$ and $P$) values at the 7 different
RH levels ranging from 23% to 94% are summarized in Table S7. Good positive
correlations existed between the water uptake and the contents of OC$_{TGA}$, OC$_{AE}$, and
dissolved minerals under high humidity conditions (Figure 2a-d). The highest
regression accuracy values obtained were $R^2 = 0.82$, $P < 0.0001$ for OC$_{TGA}$ at 84%
RH, $R^2 = 0.80$, $P = 0.001$ for OC$_{AE}$ at 94% RH, and $R^2 = 0.86$, $P = 0.0001$ for
dissolved minerals at 94% RH. However, the correlations with these BC constituents
became much weaker under low humidity conditions ($R^2 = 0.10$–0.32, $P =$
0.247–0.028 at RH = 23%). It can be concluded that the hygroscopicity of herbal BC
and soot under high humidity conditions was mainly controlled by the contents of OC
and dissolved minerals. On the other hand, the low water sorbing ability of the woody
BC was due to the very low contents of these constituents. The OC constituents in BC
contained large amounts of oxygen-containing groups and thus had very high
hygroscopicity (Xiao et al., 2013). The very strong water retention ability of dissolved
minerals in BC was understandable due to the strong hydration of mineral surfaces
and ionic constituents. No correlation was observed between the water uptake of BC





and the total organic carbon content within the whole examined RH range. Notably, a
negative correlation was observed with the EC content (Figure 2c), especially under
high humidity conditions ($R^2 = 0.54$, $P = 0.0019$ at 94% RH). Compared with the OC
in BC, the EC was comprised mainly of hydrophobic fuse aromatic hydrocarbons and
had much lower amounts of oxygen-containing groups, resulting in the very low water
sorbing ability.

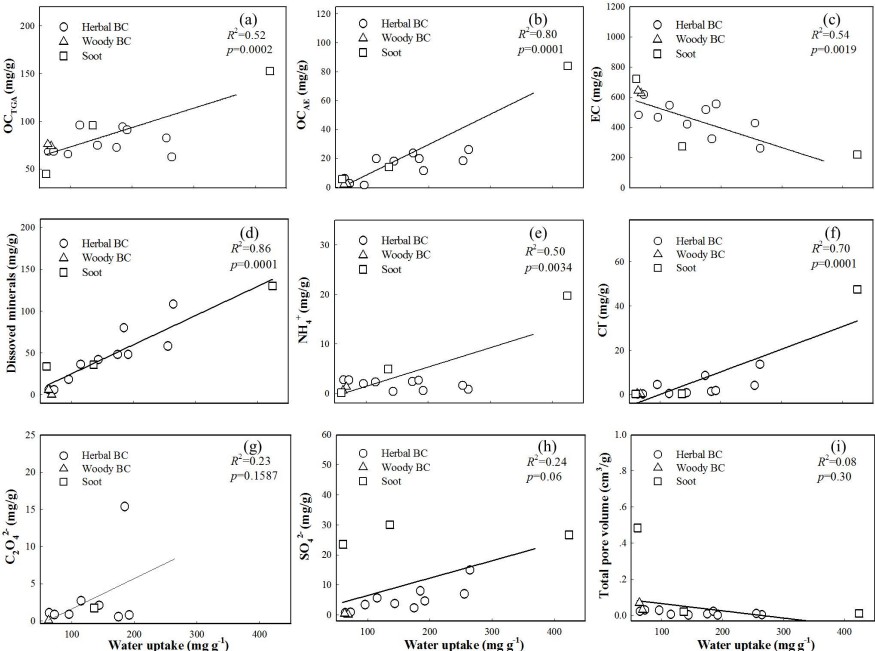

Figure 2. Relationships between equilibrium water uptake (mg g$^{-1}$) vs. compositional
and pore property parameters for the BC pool at 94% relative humidity. (a)
TGA-measured organic carbon ($OC_{TGA}$). (b) Alkali-extracted organic carbon ($OC_{AE}$).
(c) Elemental carbon (EC). (d) Dissolved minerals. (e) Ammonium ($NH_4^+$). (f)
Chloride (Cl$^-$). (g) Oxalate ($C_2O_4^{2-}$). (h) Sulfate ($SO_4^{2-}$). (i) Total porosity.
At 94% RH, relatively good positive correlations were observed with $NH_4^+$ and Cl$^-$
($R^2 = 0.50$–$0.70$, $P = 0.0001$–$0.0034$), but not with $SO_4^{2-}$ ($R^2 = 0.24$, $P = 0.06$) or


$C_2O_4^{2-}$ ($R^2 = 0.23$, $P = 0.1587$) (Figure 2e-h). No correlation ($R^2 = 0.08$, $P = 0.3$) was
observed with the total porosity of BC at 94% RH (Figure 2i). Consistently, previous
studies reported that chloride salts in biomass burning aerosols had high
hygroscopicity (Jing et al., 2017; Posfai et al., 2003). The poor correlation observed
for $SO_4^{2-}$ was ascribed to the low hygroscopicity of $CaSO_4$ and $K_2SO_4$, as evidenced
by their very high deliquescent relative humidity (96–97%) (Freney et al., 2007;
Preturlan et al., 2019). It is noteworthy that the content of $SO_4^{2-}$ positively correlated
with the contents of $Ca^{2+}$ ($R^2 = 0.74$, $P < 0.0001$) and $K^+$ ($R^2 = 0.69$, $P = 0.0008$) for
the tested BC. On the other hand, the poor correlation observed for $C_2O_4^{2-}$ was likely
due to the formation of less water-soluble salts (e.g., $K_2C_2O_4$) that might depress the
hygroscopicity (Buchholz and Mentel, 2008).
The positive correlations observed with $NH_4^+$ and $Cl^-$ at 94% RH disappeared at
23% RH. Alternatively, a relative good negative correlation ($R^2 = 0.42$, $P = 0.0095$)
with the total porosity was shown at 23% RH (Figure S7i). Similarly, a weak negative
correlation ($R^2 = 0.21$, $P = 0.083$) was shown with the EC content at 23% RH. On the
contrary, a weak positive correlation ($R^2 = 0.32$, $P = 0.028$) was observed between the
water uptake and the $OC_{TGA}$ content. It was reasonable to hypothesize that the rigid
micro- and mesoporous structures in BC were mainly formed by graphitized carbons
(EC) rather than by amorphous organic carbons (OC). Thus, the abovementioned
correlations indicated that the OC constituents played a key role in the overall water
uptake by BC under low humidity conditions.
As can be seen from the desorption isotherms in Figure 1, the herbal BC and soot
showed certain hysteresis effect (irreversible sorption), whereas the woody BC
showed no hysteresis effect. Irreversible sorption would lower the release of sorbed
water molecules from BC particles in the atmosphere when the RH changes from a



high level to a low level. The observed hysteresis effect of herbal BC and soot likely
stemmed from their relatively high contents of OC and/or dissolved minerals (such as
wheat BC and Household soot). Sorbing water molecules could cause strong and
irreversible hydration of organic acids (Petters et al., 2017) and dissolution or phase
change of minerals (Adapa et al., 2018), consequently leading to hysteresis effect due
to non-identical structures of BC between the sorption and desorption branches even
at the same RH. The negligible hysteresis effect observed on the two woody BC could
be attributed to their very low contents of OC and dissolved minerals.

The equilibrium water uptake by BC was further investigated by DRIFTS. The

spectra of representative BC (Grass BC, Red pine BC, and Household soot) at varying
RH are presented in Figure 3a-c. Figure 3d compares the water uptake at 23% RH
monitored by the integrated intensity of the O-H stretching region from 2750 to 3660
$cm^{-1}$ (Ghorai et al., 2011), along with the water uptake measured by the
multi-station gravimetric method for 8 selected BC. The identified bands of sorbed
water molecules included a combination mode of symmetric stretch around 3423 $cm^{-1}$
and asymmetric stretch stretch around 3253 $cm^{-1}$ (Gustafsson et al., 2005). The broad
feature peak centered at 2100 $cm^{-1}$ was assigned to a combined band of bending,
libration, and hindered translation modes of water, while the peak centered at 1640
$cm^{-1}$ was attributed to the bending mode of water (Ma et al., 2010). The intensities of
these peaks/bands increased with increasing RH. As assessed by the integrated
intensity of the O-H stretching region (see insets in Figure 3a-c), the water uptake by
Grass BC and Household soot increased gradually with RH from 12 to 80%; however,
the water uptake by Red pine BC rapidly reached saturation at about 28% RH, and
kept constant when the RH was further increased. With the exception of Weifu diesel
soot, the disparity pattern of water uptake by the 8 selected BC at 23% RH monitored



by DRIFTS was similar to that monitored by the multi-station gravimetric method
(Figure 3d). However, the disparities were very large between these two methods at
80% RH (Figure S8). This was probably because the saturated effect encountered in
detection of sorbed water molecules by FTIR (Gustafsson et al., 2005) became worse
under high humidity conditions. Moreover, the DRIFTS signals of sorbed water
molecules might be influenced by the distribution of sorption sites (e.g., minerals vs.
OC and exterior vs. interior), and the caused effects might lead to larger deviations
under high humidity conditions.

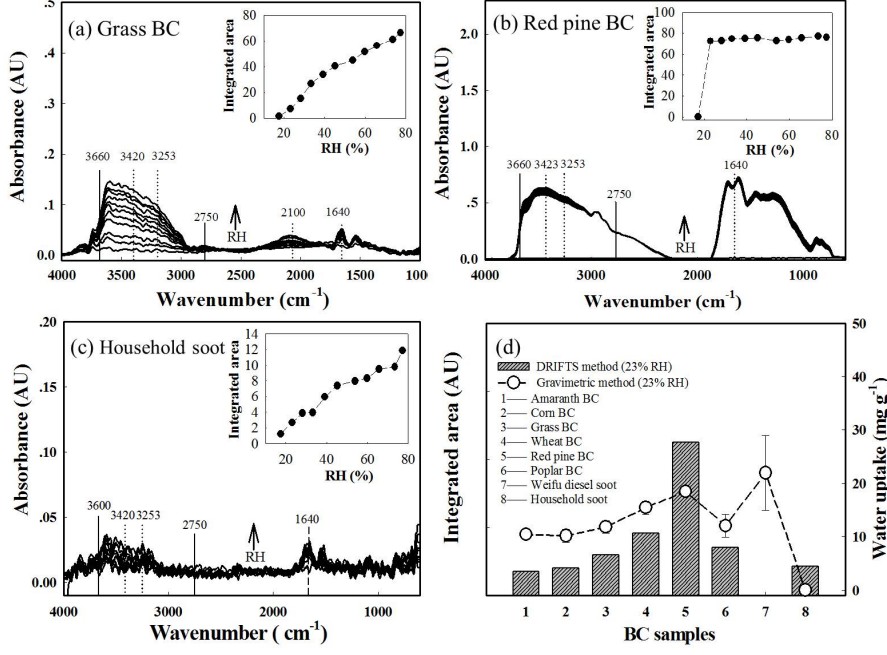


Figure 3. Diffuse reflectance infrared Fourier transform spectroscopy (DRIFTS)
characterization of equilibrium water uptake by BC. (a-c) Spectra for three
representative BC (Grass BC, Red pine BC, and Household BC) equilibrated at
varying relative humidity (RH) levels. (d) Comparison of equilibrium water uptake
measured as integrated area of O-H stretching region (2750–3660 cm$^{-1}$) between 8



selected BC at 23% relative humidity. Insets in subfigures (a-c) present water uptake
measured as integrated area of O-H stretching region against RH.
**Kinetic water uptake.** Figure 4 displays the water vapor sorption kinetics to the 15
BC at 94% RH obtained by saturated aqueous salt solutions. The sorption kinetics at
33% RH was presented in Figure S9. The two woody BC exhibited similar kinetics
curves; however, the herbal BC and soot showed very different kinetic patterns within
each group. For quantitative comparison of apparent sorption kinetics among different
BC, the data were fitted to the pseudo-first order and pseudo-second order models,
$dq_t/dt = k_1 (q_e-q_t)$ and $dq_t/dt = k_2 (q_e-q_t)^2$, respectively, where $q_t$ was the sorbed
concentration at time $t$, $q_e$ was the equilibrium sorbed concentration, and $k_1$ and $k_2$
were the pseudo-first and pseudo-second rate constants, respectively. The fitting
parameters ($q_e$ $k_1$, $k_2$) for the three selected RH levels (33, 47, and 94%) are
summarized in Table S8-S9 The pseudo-second order model ($R^2 > 0.97$) fits the data
better than the pseudo-first order model ($R^2 = 0.80–0.99$). The calculated $k_2$ differed
greatly among the BC within the herbal BC group and the soot group, but was very
close between the two woody BC. For a given BC, the $k_2$ at a lower RH level was
significantly larger than that at a higher RH level. Similar results were reported in
previous studies on sorption kinetics of water vapor to activated carbon (Ohba and
Kaneko, 2011; Ribeiro et al., 2008). Under low humidity conditions, sorption of water
vapor mainly occurs at the active, high-energy binding sites, and the sorption kinetics
is fast; alternatively, under high humidity conditions, sorption is governed by the slow
pore-filling/condensation process of water molecules within the pores of activated
carbon via formation of water clusters around the water molecules already sorbed at
the active sites (Nguyen and Bhatia, 2011; Rosas et al., 2008). Due to the small
molecular size (0.0958 × 0.151 nm, ChemDraw 3D), water molecules could well





penetrate into the micropores of BC and form water clusters via intermolecular
hydrogen bonding. The sorbing ability order of the different types of BC varied
depending on the examined RH. At 33% RH, the $k_2$ roughly followed a decreasing
order of soot ($0.5$–$5.25 \times 10^{-5}$ g mg$^{-1}$s$^{-1}$) > woody BC ($1.57$–$1.90 \times 10^{-5}$ g mg$^{-1}$s$^{-1}$) >
herbal BC ($0.34$–$2.07 \times 10^{-5}$ g mg$^{-1}$s$^{-1}$); however, no clear trend was shown for high
humidity conditions (e.g., RH = 94%), mainly resulting from the larger variances
within the herbal BC group and soot group.

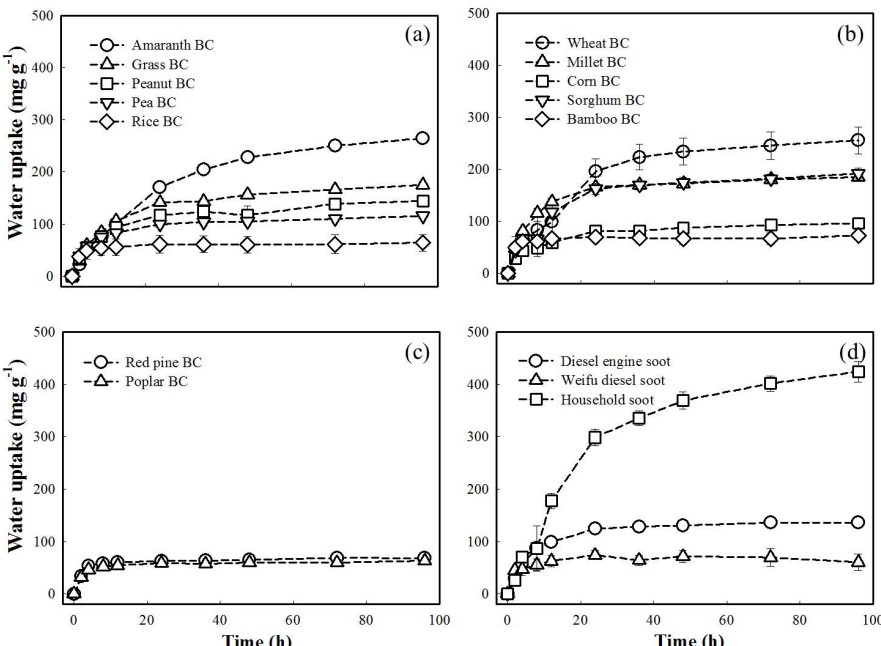


Figure 4. Sorption kinetics of water vapor plotted as water uptake (mg g$^{-1}$) vs. time (h)
at 94% relative humidity. (a) Subgroup 1 of herbal BC. (b) Subgroup 2 of herbal BC.
(c) Woody BC. (d) Soot.
Like for the equilibrium water uptake, the relationships were built between the $k_2$
and the contents of $OC_{TGA}$, $OC_{AE}$, EC, dissolved minerals, major ionic species ($NH_4^+$,
Cl$^-$, $C_2O_4^{2-}$ and $SO_4^{2-}$), and total porosity, respectively at 94% RH (Figure 5). The



regression relationships at 33% RH were presented in Figure S10. The regression
accuracy ($R^2$ and $P$) values at 33%, 47%, and 94% RH are summarized in Table S10.
At 94% RH, among the examined parameters only totally porosity was positively
correlated with the $k_2$ ($R^2 = 0.82$, $P < 0.0001$). This correlation disappeared under
low and medium humidity conditions. The strong positive correlation between $k_2$ and
total porosity at high RH can be well explained by the pore-filling/condensation
mechanism. A similar mechanism has been previously proposed to account for the
positive correlation observed between the water vapor sorption kinetics and the
porosity of activated carbon under high humidity conditions (Nakamura et al., 2010;
Velasco et al., 2016). At 33% RH, relatively good positive correlations were observed
with the contents of $OC_{TGA}$ ($R^2 = 0.47$, $P = 0.0046$), $OC_{AE}$ ($R^2 = 0.44$, $P = 0.007$),
$NH_4^+$ ($R^2 = 0.77$, $P < 0.0001$), and $Cl^-$ ($R^2 = 0.60$, $P = 0.0007$), but not with $SO4^{2-}$
($R^2 = 0.11$, $P = 0.2286$) or dissolved minerals ($R^2 = 0.08$, $P = 0.31$). The positive
correlations with these constituents were not shown under medium and high humidity
conditions. Thus, the constituents of OC and $NH_4^+$- and $Cl^-$-salts likely provided the
primary high affinity, active sites for sorption of water vapor under low humidity
conditions.



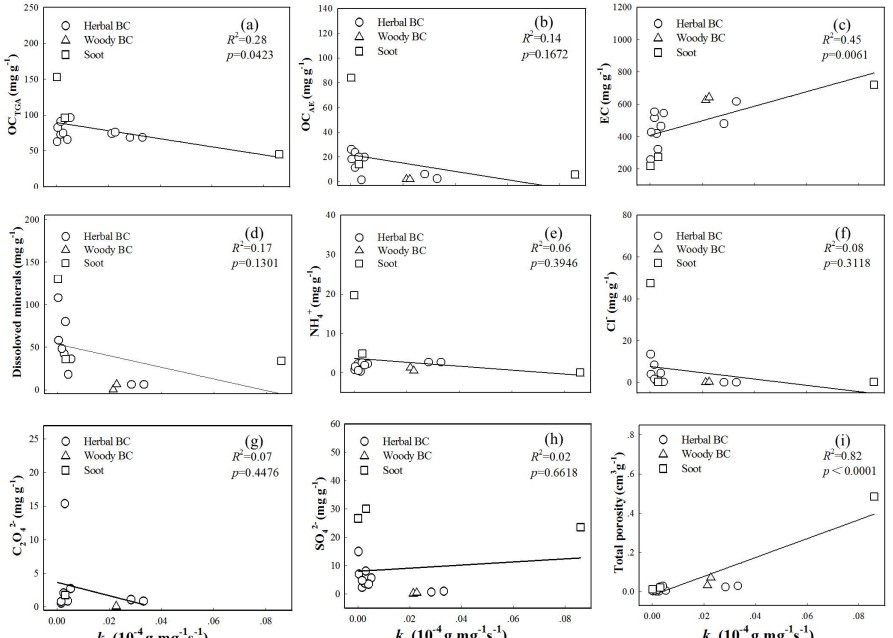

Figure 5. Relationships between pseudo-second water uptake rate constant ($k_2$) (g $mg^{-1}s^{-1}$) vs. compositional and pore property parameters for the BC pool at 94% relative humidity. (a) TGA-measured organic carbon ($OC_{TGA}$). (b) Alkali-extracted organic carbon ($OC_{AE}$). (c) Elemental carbon (EC). (d) Dissolved minerals. (e) Ammonium ($NH_4^+$). (f) Chloride ($Cl^-$). (g) Oxalate ($C_2O_4^{2-}$). (h) Sulfate ($SO_4^{2-}$). (i) Total porosity.

**4. Conclusion**

The hygroscopic properties of 15 different BC (herbal, woody, and soot) were systematically investigated using gravimetric method and DRIFTS. The mechanisms and factors controlling the equilibrium and kinetic water uptake differed among the types of BC and depended heavily on the humidity conditions. Linear correlation analyses indicated that the equilibrium water uptake by the tested BC pool positively correlated to the contents of OC ($OC_{TGA}$ and $OC_{AE}$), dissolved minerals, and $NH_4^+$- and $Cl^-$-salts under high humidity conditions, and weakly to the contents of OC only





under low humidity conditions. By contrast, negative correlations were observed
between the equilibrium water uptake and the EC content. The low water uptake by
the woody BC could be attributed to the very low contents of OC and dissolved
minerals. Thus, the equilibrium water uptake by BC was mainly controlled by the
hygroscopic constituents of OC and dissolved minerals/salts. The kinetic water uptake
by the BC could be well described by the pseudo-second order kinetic model. The
calculated rate constant ($k_2$) positively correlated to the contents $OC_{TGA}$, $OC_{AE}$, and
$NH_4^+$- and $Cl^-$-salts under low humidity conditions, and to the total porosity only
under high humidity conditions. The fast water uptake kinetics under low humidity
conditions was attributed to the binding to high affinity, active sites (OC and salts),
whereas the slow water uptake kinetics under high humidity conditions was attributed
to pore-filling/condensation of water molecules within the micro- and mesopores of
BC. This study highlights that the hygroscopic properties of BC rely on compositional
and structural properties of BC as well as humidity conditions.
**Author contributions.** DZ provided the original idea and prepared the paper with
contributions from all co-authors. MW and YC designed and conducted the research,
HF, XQ were involved in the development of the analysis methods. BL, ST reviewed
the written document.
**Competing interests.** The authors declare that they have no conflict of interest.
**Acknowledgments.** This work was supported by the National Natural Science
Foundation of China (Grants 21777002, 21920102002, and 41991331).
**Appendix A.**
Detailed characterization results of the different BC can be found in Table S1-S5.
Table S6 lists accuracy ($R^2$ and $P$) values for regression on equilibrium water uptake
against different variables. Table S7-S8 presents pseudo-first/second order kinetic



model fitting parameters. Table S9 lists accuracy ($R^2$ and $P$) values for regression on
$k_2$ against different variables. Figure S1-S4 displays spectroscopic characterization of
different BC. Figure S5 displays sorption isotherms at selected humidity obtained by
using saturated aqueous salt solutions. Figure S6 compares equilibrium water uptake
measured by the two different gravimetric methods. Figure S7 displays relationships
between equilibrium water uptake and different variables at 23% RH. Figure S8
compares equilibrium water uptake measured by DRIFTS and gravimetric method at
high RH. Figure S9 displays sorption kinetics of water uptake at 33% RH. Figure S10
displays relationships between $k_2$ and different variables at 33% RH.



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
