# Peer review of "An investigation on hygroscopic properties of 15 black carbon (BC) from different carbon sources: Roles of organic and inorganic components"

_Atmospheric Chemistry and Physics, 2020_

## Referee Comment (RC1) · Anonymous Referee #1 · 21 Apr 2020

The manuscript describes the hygroscopic properties of 15 black carbon (BC) from different carbon sources using gravimetric method and DRIFTS. The authors found different BC had different mechanisms and controlling factors for equilibrium and kinetic water uptake, which mainly depended on humidity conditions. The 15 BC were characterized by many kinds of methods. The contents and constituents of OC and minerals/salts have great impact on the hygroscopic properties by this study. Therefore, it is recommended to publish on ACP after revision. Some and more detailed comments are included below:

1. In the experimental part, the author should give more information of BC sources.

2. The particles should not pass through 100-mesh sieve (0.15 mm) only once, what is the particle size of the prepared BC particles? Different specific surface areas with different particle sizes will greatly affect the experimental results.

3. Lines 212-214: Why choose 30 minutes as the equilibrium time? Has the author conducted a series of equilibrium time gradient experiments to choose the preparing time?

4. Line 276: the "Si" should be changed to "Si-".

5. Line 313: The data in Figure 1 must be not done only once experiment, the error bar should be added.

6. Lines 361-363: Expect the water-soluble salts, even the insoluble compound (such as H2C2O4) could become water-soluble compound. The author can explain the phenomenon in this way.

7. Line 411: The absorption peaks of water were not unchanged, so the authors can refer to the spectra and previous study to ensure the position of those peaks. In figure 3, the signed peaks around 3600 cm-1 are different from the strong peaks.

8. Line 411: In figure 3(a), 1000 in the abscissa is shown as 100 due to layout style.

9. Lines 496-500: The conclusions can be more credible if the author can add some photos of BC particles when discussing the influence of BC's micro-control and mesoporous.

---

## Referee Comment (RC2) · Anonymous Referee #2 · 5 May 2020

**General comments**

This paper reports on the hygroscopic properties of 15 different types of black carbon (BC) aerosol particles. Both equilibrium and kinetics of water uptake were measured with various methods. The results indicate a fairly wide variation of hygroscopicity amongst the BC types. These variations correlate with the types of dissolved minerals, organic carbon, and soluble ions present with the black carbon.

Overall the paper is fairly well-written, with some suggested corrections listed below. A few concerns listed below should be addressed prior to publication.

[Figure]

**Specific comments**

Throughout the paper, be more specific and clear when referring to "black carbon". For example, in the title of the paper, I think you mean to say something like "15 black carbon types". Adding the word "types" at other points in the manuscript are necessary as well (e.g. line 420). You should also clearly define "soot". Why is it not diesel BC or herbal/woody soot?

Along the lines of the previous comment, I don't think that Line 61 is a completely accurate definition of BC. You reference Bond et al., 2013 in Line 43, so I would assume you would follow their definition. Bond et al., 2013 defines BC as being refractory, insoluble, and consisting of an aggregate of small carbon spherules (among other attributes). Thus, does "BC" really encompass the salts and minerals you mention? (In my opinion, no.)

Again, going back to my first comment, be more clear and specific in your language. You are measuring the hygroscopicity of BC-containing particles from these various sources and your results show how hygroscopicity changes with different impurities in the BC.

Some more details on the 3 soot types should be provided. How long was the household soot on the walls of the oven before collection? How much did this soot have contact with air? In other words, what was the likelihood that other gases not related to the actual combustion were adsorbed to this soot? Please also describe the differences in diesel soot - what do you expect the differences to be simply resulting from collection on a filter versus the walls of the tailpipe? Does the diesel engine have a particulate filter installed? I presume that "# 5" and "# 0" refer to different types of diesel fuel? Please explain.

Line 180 - Please comment on how drying the samples would effect interpretation of your results for ambient BC aerosol.

Line 183 - How do you know?

Line 227-228 - I do not understand this sentence.

Figure 1 - What are the error bars/uncertainty on each datapoint? You talk about the hysteresis in the plots so it is important to establish that the adsorption and desorption curves are statistically different from each other. Also, the way these are plotted, some of the hysteresis is hard to see anyways. On Line 374, not all BC/soot curves are obviously hysteretic, so please specify here.

Line 327 - The results at 84% humidity are not shown anywhere in the paper or supplemental. Please add.

Line 330 - Is the weak correlation simply due to lower signals?

Line 334 - The OC constituents in which BC type? Can you point to specifics in Table 1 at this point in the paper as well?

Figure 2 - Rather than look at each constituent individually, what about a multi-factor approach, similar to Positive Matrix Factorization (PMF)? If it is always true, for example, that several of your compositional properties are always correlated, then separating them as you do in Figure 2 does not actually reveal any new information. You have plenty of data in this study to feed into a multi-factor analysis, and I think you may learn quite a bit from that type of analysis.

Line 369-373 - This seems like a reasonable hypothesis, but the way this section is worded makes it sound like you came up with the hypothesis to fit the results rather than vice-versa. You could cite some literature here to strengthen your arguments and show how your data supports this claim.

Figure 3 - Please use the same x-axis limits on parts a, b, and c.

Line 429-430 - Please explain the physical meaning behind a second-order model fitting the data better than a first-order model. Why is this important?

Line 458 - Figure 5 shows that EC is also positively correlated, not just total porosity (spelling error there too).

Line 496-500 - Why aren't there active sites at high humidity conditions?

**Technical corrections**

Line 25 - I don't understand the use of "BC pool" here.

Line 42 - Is recalcitrant the right word to use here?

There are several uses of "this" as the subject/noun in your sentences. Please be more clear in your writing.

There are a few other typos, missing "-s" or "the", missing spaces in captions, etc., throughout the paper. Give it another careful proofread before publication. Be consistent on whether using past or present tense.

---

## Author Comment (AC1) · 23 May 2020

We have one-by-one responded the Referee's comments. Please see attached zip.

Please also note the supplement to this comment:
https://www.atmos-chem-phys-discuss.net/acp-2020-110/acp-2020-110-AC1-supplement.zip

---

## Author Comment (AC2) · 23 May 2020

We have one-by-one responded the Referee's comments. Please see attached PDF.

Please also note the supplement to this comment:
https://www.atmos-chem-phys-discuss.net/acp-2020-110/acp-2020-110-AC2-supplement.zip

---

## Author Response (AR1)

**Comment**

The manuscript describes the hygroscopic properties of 15 black carbon (BC) from different carbon sources using gravimetric method and DRIFTS. The authors found different BC had different mechanisms and controlling factors for equilibrium and kinetic water uptake, which mainly depended on humidity conditions. The 15 BC were characterized by many kinds of methods. The contents and constituents of OC and minerals/salts have great impact on the hygroscopic properties by this study. Therefore, it is recommended to publish on ACP after revision. Some and more detailed comments are included below:

**Reply**

We appreciate the reviewer's positive comments on the paper.

**Comment**

1. In the experimental part, the author should give more information of BC sources.

**Reply**

The geographic locations for collecting the biomass used in preparation of herb and woody BCPs were given in the revised paper.

More information on the two diesel soots and the household soot were also given (shown as below).

A total of 15 BCPs were tested, including 10 herb-derived BCPs, 2 wood-derived BCPs, and 3 soot-type BCPs. The herbs used for preparation of herbal BCPs were amaranth, peanuts, pea, grass, rice, wheat, corn, millet, sorghum (Nantong, Jiangsu Province, China), and bamboo (Lishui, Zhejiang Province, China), and the woods used for preparation of woody BCPs were red pine and poplar (Lishui, Zhejiang Province, China). The dried and dehydrated biomass was pulverized into a fine powder using a high-speed pulverizer (FW 100, Tianjin Taisite Instrument, China), and pyrolyzed in a muffle furnace under an oxygen-limited conditions. The oven temperature was programmed to increase from 20 to 400 °C in 2 h and maintained at 400 °C for 3 h. The Weifu diesel soot was purchased from Wuxi Weifu Automotive Diesel System Co., Ltd. (Jiangsu Province, China). According to the information provided by the manufacturer, the soot was produced under laboratory conditions by burning diesel (type # 5, China) at 1000  $\,^{\circ}$ C and was collected by a diesel particulate filter (NGK-6000YE) from the exhaust stream at a carbon deposition temperature of 250 °C. The diesel engine soot was collected in outdoor conditions from the freshly discharged exhaust particles on the tailpipe of a diesel truck (Dongfeng, CY4100, 2015, China) equipped with a diesel engine (CY4100Q, 3.7 L, Diesel # 0) and a diesel particulate filter (BST-5L-QCD). The household soot was collected on the inner wall

of a stove chamber (Linkou County, Heilongjiang Province, China) 1 hour later after burning of coal and wood for winter cooking and heating under limited oxygen exposure. (Page 6, lines 116-136)

**Comment**

2. The particles should not pass through 100-mesh sieve (0.15 mm) only once, what is the particle size of the prepared BC particles? Different specific surface areas with different particle sizes will greatly affect the experimental results.

**Reply**

Grounding and sieving are often applied as a standard protocol to obtain fine BCPs particles with relatively homogeneous sizes for experimental reproducibility. Please note the pretreatment would not change the compositional and structural properties of the tested BCPs.

**Comment**

3. Lines 212-214: Why choose 30 minutes as the equilibrium time? Has the author conducted a series of equilibrium time gradient experiments to choose the preparing time?

**Reply**

We understand the reviewer's concern. We measured the adsorption kinetics (see below) and found that 30 minutes were long enough to reach adsorption equilibrium. Please refer to the kinetic data below.

Figure R1. Equilibrium water uptake by different BCPs at 94% relative humidity measured as integrated area of O-H stretching region  $(2750-3660 \text{ cm}^{-1})$  vs. time (h).

**Comment**

4. Line 276: the "Si" should be changed to "Si-".

Reply Revised. Thanks.

**Comment**

5. *Line 313: The data in Figure 1 must be not done only once experiment, the error bar should be added.*

**Reply**

Figure 1 presents sorption and desorption isotherms of water vapor to different BCPs. It is common to see that desorption isotherms of water vapor to porous media are conducted with single-point (not in replicate). With the exception of the data presented in Figure 1, all other sorption isotherm experiments were conducted in duplicate, including those for linear regression analysis. Revisions on the captions of Figures 1, 3, and 4 were made to clarify whether the data were collected in duplicate or not. The data of equilibrium water uptake for linear regression were obtained by saturated aqueous salt solutions and collected in duplicate (Figure S5). The error bars for most of the data points are smaller than the symbols.

**Comment**

6. Lines 361-363: Expect the water-soluble salts, even the insoluble compound (such as H2C2O4) could become water-soluble compound. The author can explain the phenomenon in this way.

**Reply**

Thanks for the suggestion. The insoluble compounds are indeed  $CaC_2O_4$  and/or  $H_2C_2O_4$ . Please refer to the follow revision.

On the other hand, the poor correlation observed for  $C_2O_4^{2-}$  was likely due to the formation of less water-soluble compounds (e.g.,  $CaC_2O_4$  and  $H_2C_2O_4$ ) that might depress the hygroscopicity (Buchholz and Mentel, 2008). (Page 19, lines 389-391)

**Comment**

7. Line 411: The absorption peaks of water were not unchanged, so the authors can refer to the spectra and previous study to ensure the position of those peaks. In figure 3, the signed peaks around 3600 cm-1 are different from the strong peaks.

**Reply**

Thanks for the suggestion. Please note the wavelength of  $3600 \text{ cm}^{-1}$  was a clerical error. It should be  $3660 \text{ cm}^{-1}$ . The water uptake was monitored by the integrated intensity of the O-H stretching region from 2750 to  $3660 \text{ cm}^{-1}$  according to previous study.

**Comment**

8. Line 411: In figure 3(a), 1000 in the abscissa is shown as 100 due to layout style.

**Reply**

Thanks for pointing this out. Revised.

**Comment**

9. Lines 496-500: The conclusions can be more credible if the author can add some photos of BC particles when discussing the influence of BC's micro-control and mesoporous.

**Reply**

Thanks for the suggestion. We indeed measured the microporosity and the mesoporosity. The data were presented in Table S4 for better clarity.

**Anonymous Referee #2**

Received and published: 5 May 2020

**Comment**

**General comments**

This paper reports on the hygroscopic properties of 15 different types of black carbon (BC) aerosol particles. Both equilibrium and kinetics of water uptake were measured with various methods. The results indicate a fairly wide variation of hygroscopicity amongst the BC types. These variations correlate with the types of dissolved minerals, organic carbon, and soluble ions present with the black carbon. Overall the paper is fairly well-written, with some suggested corrections listed below. A few concerns listed below should be addressed prior to publication.

**Reply**

We appreciate the reviewer's positive comments.

**Comment**

**Specific comments**

Throughout the paper, be more specific and clear when referring to "black carbon". For example, in the title of the paper, I think you mean to say something like "15 black carbon types". Adding the word "types" at other points in the manuscript are necessary as well (e.g. line 420). You should also clearly define "soot". Why is it not diesel BC or herbal/woody soot?

**Reply**

We appreciate the reviewer's suggestion. The term "black carbon-containing particles (BCPs)" was adopted from the literature to better describe the samples tested in this study.

Following the reviewer's suggestion, we explicitly described the structural and morphological properties of soot (shown as below).

Compared with char and charcoal, soot (another type of BCPs) produced from fossil fuel combustion is comprised of more regular shaped, chain-like agglomerates of primary particles, which consist of perturbed graphitic layers oriented concentrically in an onion-like fashion (Nienow and Roberts, 2006). (Page 5, lines 99-103)

**Comment**

Along the lines of the previous comment, I don't think that Line 61 is a completely accurate definition of BC. You reference Bond et al., 2013 in Line 43, so I would assume you would follow their definition. Bond et al., 2013 defines BC as being refractory, insoluble, and consisting of an aggregate of small carbon spherules (among other attributes). Thus, does "BC" really encompass the salts and minerals you mention? (In my opinion, no.)

**Reply**

We understand the reviewer's concern. Please refer to the above reply. A more accurate definition of black carbon was provided according to Bond et al., 2013 (shown as below).

Black carbon (BC) refers to a collective term of refractory and insoluble aggregates of small carbon spherules generated from incomplete combustion of biomass and fossil fuels (Bond et al., 2013). (Page 3, lines 44-46)

**Comment**

Again, going back to my first comment, be more clear and specific in your language. You are measuring the hygroscopicity of BC-containing particles from these various sources and your results show how hygroscopicity changes with different impurities in the BC. Some more details on the 3 soot types should be provided. How long was the household soot on the walls of the oven before collection? How much did this soot have contact with air? In other words, what was the likelihood that other gases not related to the actual combustion were adsorbed to this soot? Please also describe the differences in diesel soot - what do you expect the differences to be simply resulting from collection on a filter versus the walls of the tailpipe? Does the diesel engine have a particulate filter installed? I presume that "# 5" and "# 0" refer to different types of diesel fuel? Please explain.

**Reply**

We appreciate the reviewer's suggestion. Please note that the BCPs were dried at 70  $^{\circ}$ C under vacuum for 12 h to remove residual water and other adsorbed gasses prior to testing of water vapor adsorption. The major difference among the three soot-type BCPs was that they were produced under different conditions. Some revisions were made to better describe the samples tested in the study (shown as below).

A total of 15 BCPs were tested, including 10 herb-derived BCPs, 2 wood-derived BCPs, and 3 soot-type BCPs. The herbs used for preparation of herbal BCPs were amaranth, peanuts, pea, grass, rice, wheat, corn, millet, sorghum (Nantong, Jiangsu Province, China), and bamboo (Lishui, Zhejiang Province, China), and the woods used for preparation of woody BCPs were red pine and poplar (Lishui, Zhejiang Province, China). The dried and dehydrated biomass was pulverized into a fine powder using a high-speed pulverizer (FW 100, Tianjin Taisite Instrument, China), and pyrolyzed in a muffle furnace under an oxygen-limited conditions. The oven temperature was programmed to increase from 20 to 400  $^{\circ}$ C in 2 h and maintained at 400  $^{\circ}$ C for 3 h. The Weifu diesel soot was purchased from Wuxi Weifu Automotive Diesel System Co., Ltd. (Jiangsu Province, China). According to the information provided by the manufacturer, the soot was produced under laboratory conditions by burning diesel (type # 5, China) at 1000  $^{\circ}$ C and was collected by a diesel particulate

filter (NGK-6000YE) from the exhaust stream at a carbon deposition temperature of 250 °C. The diesel engine soot was collected in outdoor conditions from the freshly discharged exhaust particles on the tailpipe of a diesel truck (Dongfeng, CY4100, 2015, China) equipped with a diesel engine (CY4100Q, 3.7 L, Diesel # 0) and a diesel particulate filter (BST-5L-QCD). The household soot was collected on the inner wall of a stove chamber (Linkou County, Heilongjiang Province, China) 1 hour later after burning of coal and wood for winter cooking and heating under limited oxygen exposure. (Page 6, lines 116-136)

**Comment**

Line 180 - Please comment on how drying the samples would effect interpretation of your results for ambient BC aerosol.

**Reply**

Thanks for the suggestion. Please refer to the revision made as follows.

Prior to testing, the BCPs (about 10 mg) were dried at 70  $\,^{\circ}$ C under vacuum for 12 h to remove residual water vapor and set a baseline for comparison of water vapor adsorption behaviors among different samples. (Page 9, lines191-194)

Comment Line 183 - How do you know?

**Reply**

The amount of water sorbed to the sample tube was measured and showed to be negligible (Page 9, line 194-197).

*Comment Line* 227-228 - *I do not understand this sentence*.

Reply

Please refer to the following revision.

Similarly, the surfaces of all the 15 BCPs were dominated by C and O; however, the differences of the surface C, O compositions among the 15 BCPs were much smaller compared to the bulk C, O compositions. (Page 10-11, lines 239-241)

**Comment**

Figure 1 - What are the error bars/uncertainty on each datapoint? You talk about the hysteresis in the plots so it is important to establish that the adsorption and desorption curves are statistically different from each other. Also, the way these are plotted, some of the hysteresis is hard to see anyways. On Line 374, not all BC/soot curves are obviously hysteretic, so please specify here.

**Reply**

Unlike other experimental data, the data presented in Figure 1 were collected with single-point. It is common to see that desorption isotherms of water vapor to porous media are conducted with single-point (not in replicate). The data of equilibrium water uptake for linear regression were obtained by saturated aqueous salt solutions and collected in duplicate (Figure S5). The error bars for most of the data points are smaller than the symbols.

In the revised version, dash color lines were used to increase discrimination and clarity.

In fact, only some of the herb BCPs and soot samples showed strong hysteresis effect. Please refer to the following revision.

As can be seen from the sorption-desorption isotherms in Figure 1, some herbal BCPs (Amarance, Peanuts, Wheat, Millet, Pea, and Sorghum) and soot (Diesel engine soot and Household soot) showed strong hysteresis effect (irreversible sorption), whereas none of the woody BCPs showed hysteresis effect. (Page 19, lines 402-405)

**Comment**

Line 327 - The results at 84% humidity are not shown anywhere in the paper or supplemental. Please add.

**Reply**

Thanks for pointing this out. Please refer to the revision made as follows for better clarity.

The regression accuracy ( $R^2$  and P) values at the 7 different RH levels (23, 33, 43, 47, 75, 84, 94%) are summarized in Table S8. (Page 16, lines 341-343)

**Comment**

*Line 330 - Is the weak correlation simply due to lower signals?*

**Reply**

Not really. The data of equilibrium water uptake could be accurately measured even at low humidity levels. As seen from Figure S5, most of the error bars are smaller than the symbols.

**Comment**

*Line 334 - The OC constituents in which BC type? Can you point to specifics in Table 1 at this point in the paper as well?*

**Reply**

Thanks for pointing this out. Please refer to the revision as follows.

As indicated by the elemental analysis results (Table S2), the OC constituents in the three representative BCPs (Grass, Red pine, and Household soot) contained large amounts of oxygen-containing groups, which expectedly had very high hygroscopicity (Xiao et al., 2013). (Page 17, lines 353-356)

**Comment**

Figure 2 - Rather than look at each constituent individually, what about a multi-factor approach, similar to Positive Matrix Factorization (PMF)? If it is always true, for example, that several of your compositional properties are always correlated, then separating them as you do in Figure 2 does not actually reveal any new information. You have plenty of data in this study to feed into a multi-factor analysis, and I think you may learn quite a bit from that type of analysis.

**Reply**

We appreciate the reviewer's suggestion. Indeed, a better correlation relationship could be obtained by binary factor linear regression. Please refer to the following revision.

The above one-factor linear correlation analysis indicted that at high RH the hygroscopicity of BCPs was dominated by OC and dissolved minerals. Consistently, a better positive correlation ( $R^2 = 0.90$ , P < 0.0001) could be obtained between the equilibrium water uptake and the contents of OCTGA and dissolved mineral by binary linear regression for the tested group of BCPs (Figure S8). Interestingly, the correlation relationship became poorer as the RH level gradually increased, and the worst correlation was shown at 23% RH ( $R^2 = 0.21$ , P = 0.028). The results suggested that the hygroscopicity of BCPs at low RH was controlled by different factors. (Page 17-18, lines 365-372)